# Inhibitory Mechanisms of Lekethromycin in Dog Liver Cytochrome P450 Enzymes Based on UPLC-MS/MS Cocktail Method

**DOI:** 10.3390/molecules28207193

**Published:** 2023-10-20

**Authors:** Pan Sun, Yuying Cao, Jicheng Qiu, Jingyuan Kong, Suxia Zhang, Xingyuan Cao

**Affiliations:** 1Department of Veterinary Pharmacology and Toxicology, College of Veterinary Medicine, China Agricultural University, Beijing 100193, China; pan.sun@ugent.be (P.S.); cyy@cau.edu.cn (Y.C.); qiujicheng2015@163.com (J.Q.); jingyuankong@outlook.com (J.K.); suxia@cau.edu.cn (S.Z.); 2Laboratory of Quality & Safety Risk Assessment for Animal Products on Chemical Hazards (Beijing), Ministry of Agriculture and Rural Affairs of the People’s Republic of China, Beijing 100193, China; 3Key Laboratory of Detection for Veterinary Drug Residues and Illegal Additives, Ministry of Agriculture and Rural Affairs of the People’s Republic of China, Beijing 100193, China

**Keywords:** lekethromycin, CYP450 enzymes, drug–drug interaction, dog liver microsomes

## Abstract

Lekethromycin (LKMS) is a synthetic macrolide compound derivative intended for use as a veterinary medicine. Since there have been no in vitro studies evaluating its potential for drug–drug interactions related to cytochrome P450 (CYP450) enzymes, the effect of the inhibitory mechanisms of LKMS on CYP450 enzymes is still unclear. Thus, this study aimed to evaluate the inhibitory effects of LKMS on dog CYP450 enzymes. A cocktail approach using ultra-performance liquid chromatography–tandem mass spectrometry was conducted to investigate the inhibitory effect of LKMS on canine CYP450 enzymes. Typical probe substrates of phenacetin, coumarin, bupropion, tolbutamide, dextromethorphan, chlorzoxazone, and testosterone were used for CYP1A2, CYP2A6, CYP2B6, CYP2C9, CYP2D6, CYP2E1, and CYP3A4, respectively. This study showed that LKMS might not be a time-dependent inhibitor. LKMS inhibited CYP2A6, CYP2B6, and CYP2D6 via mixed inhibition. LKMS exhibited mixed-type inhibition against the activity of CYP2A6 with an inhibition constant (K_i_) value of 135.6 μΜ. LKMS inhibited CYP2B6 in a mixed way, with K_i_ values of 59.44 μM. A phenotyping study based on an inhibition assay indicated that CYP2D6 contributes to the biotransformation of LKMS. A mixed inhibition of CYP2D6 with K_i_ values of 64.87 μM was also observed. Given that this study was performed in vitro, further in vivo studies should be conducted to identify the interaction between LKMS and canine CYP450 enzymes to provide data support for the clinical application of LKMS and the avoidance of adverse interactions between other drugs.

## 1. Introduction

Given the high research and development costs, long development process, and high failure rate of new drugs, the related problems of drug–drug interactions (DDIs) should be solved. Lekethromycin (LKMS), a derivative synthetic novel macrolide compound, has been used in veterinary medicine to treat respiratory disease. Pharmacokinetic and distribution studies have been completed on healthy and infected rats with different administration methods and doses, respectively [1,2,3], suggesting that LKMS exhibits widely distributed effects and has a high plasma protein binding rate and low clearance. A high plasma protein binding rate typically means that there is less drug in the free state in plasma, and a DDI should be considered when co-administrating [4]. Absorption, distribution, metabolism, and excretion (ADME) analysis is necessary to evaluate safety and risks when exposing dogs to drugs. Given the complexity of diseases and drug resistance, combining chemical drugs is widely used to treat infectious diseases. Adverse drug interactions occur in combination therapies because of the large number and variety of drugs. DDI studies can evaluate pre-selected medication regimens and develop the safest and quality-controlled dosing regimens in clinical treatments. Pharmacokinetic interactions assess the effect of DDIs by comparing the ADME value of the test drug with and without a perpetrator. In January 2020, The United States Food and Drug Administration (FDA) issued two guidance standards entitled “Clinical Drug Interaction Studies-Guidance for Industry on Cytochrome P450 Enzyme and Transporter-Mediated Drug Interactions” and “In vitro Drug Interaction Studies-Cytochrome P450 Enzyme and Transporter-Mediated Drug Interactions”, which guide studies on drug interactions based on metabolic enzymes and transporters [5,6,7,8]. DDI methods include probe drug cocktail methods, liver microsome and hepatocyte models, static models, physiologically based pharmacokinetic models, machine learning models, in vivo comparative efficacy studies, and in vitro static and dynamic tests [8].

Our study focused on the in vitro interactions of hepatic microsomal methods. Cytochrome P450 enzyme (CYP450) is a metabolic enzyme that participates in more than 90% of reported enzymatic reactions [9]. CYP450 plays an important role in drug detoxification, cell metabolism, and balance and influences a medicine’s effect and safety. The liver, intestine, lung, kidney, brain, heart, adrenal, gonads, and most other organs have the capability for biotransformation through the CYP450 enzyme [10,11,12,13]. Among these, the liver is the major organ involved in CYP450-mediated metabolism. Specifically, CYP450 is involved in the phase I reactions of drug degradation in the liver [14]. Isomers of the CYP1-3 family are responsible for approximately 80% of the metabolism of clinical drugs [15]. In total, 37% of medicines were metabolized by CYP3A4/5, 15% by CYP2D6, and 27% by CYP2C9 and CYP2C19 [16,17]. Notably, CYP450 can be inhibited or induced by xenobiotics, resulting in DDIs that might cause unexpected adverse effects or treatment failure [18]. Numerous in vitro cocktails have been developed for detecting direct DDIs and screening time-dependent inhibition. Therefore, the metabolic interactions between drugs should be paid more attention to regarding the effect of medicine on the activity of CYP450.

Macrolide antibiotics have been widely used on livestock. The effect of macrolides on CYP450 can contribute to DDIs, such as erythromycin and its precursor drugs, which have been shown to inhibit the CYP3A4 enzyme, reducing the clearance of other drugs catalyzed by the CYP3A4 enzyme [19,20,21,22,23]. Clarithromycin has been found to inhibit the CYP3A5 enzyme when exposed to high concentrations but has no obvious inhibition at low concentrations [24]. Intriguingly, azithromycin exhibits a weak inhibitory effect on CYP3A4 compared with other macrolide drugs [25]. Telithromycin has been shown to decrease the activity of CYP1A2 and CYP3A2 [26,27,28]. However, rokitamycin does not affect CYP1A2 or CYP2C9. The metabolic processing of drugs metabolized by CYP450 can increase or decrease the blood concentration, which can induce adverse reactions; hence, medicine combinations should be given special attention.

In the early stage of drug development, when measuring the in vitro half-maximal inhibitory concentration (IC_50_) to study the inhibitory effect of a drug on the CYP450 enzyme, the possibility of drug interaction with other drugs can be calculated theoretically, which can be predicted by creating a simple in vitro model [29,30]. The preliminary constructed in vitro cocktail provides data support for the subsequent in vivo cocktail establishment. Simulations using well-established in vitro models and actual in vivo situations sometimes have some differences, and cocktails can assess the interactions between metabolic enzymes in vivo, improving the accuracy of in vitro prediction [31,32]. It is important to understand the metabolic process both in vitro and in vivo and to obtain pharmacokinetic parameters for preclinical research and clinical applications [33]. In vitro, CYP450-related metabolic studies are considered cost-effective in predicting potential drug interactions and are one of the main ways of predicting drug interactions in drug development, but it is important to recommend a reliable and durable liver microsomal incubation system for the reliability of experimental results. Thus, it is critical to select enzyme substrates and rationally optimize incubation system conditions [34]. In this study, a cocktail method based on ultra-performance liquid chromatography–tandem mass spectrometry (UPLC-MS/MS) was used to detect the in vitro effect of LKMS on CYP450 enzymes and the mechanism of drug interaction [35,36] to provide data support for the clinical application of LKMS and the avoidance of adverse interactions between other drugs.

## 2. Results

### 2.1. Methods Validation

A rapid and highly sensitive method was validated for the quantification of probe substrates in dog liver microsomes (DLMs) following the guidelines for bioanalytical method validation [37]. The acetonitrile protein precipitation method was applied to develop a sensitive UPLC-MS/MS method for determining the specific metabolites of each CYP450 probe substrate [38,39,40,41]. As shown in Figure 1, there were no interfering peaks detected in the blank sample at the retention time of the target compounds and internal standard. The limit of detection and the limit of quantification of the metabolites of the CYP450 probe substrates were 5 ng/mL and 10 ng/mL, respectively. Calibration curves were constructed for each compound and showed good linearity (R^2^ > 0.99). The accuracy and precision are shown in Table 1. The results of intra-day and inter-day precision were within 12.14% and an accuracy of 95.53–109.42%.

### 2.2. Enzyme Kinetic Study

It is important to use the right probe substrates, and the most common in vitro probes include phenacetin (PH) for CYP1A2, coumarin (CAN) for CYP2A6, bupropion (BP) for CYP2B6, tolbutamide (TBD) for CYP2C9, dextromethorphan (DEX) for CYP2D6, chlorzoxazone (CLN) for CYP2E1, and testosterone (TS) for CYP3A4 [42,43,44,45,46,47]. The activity of CYP450 isoforms was evaluated via correspondence based on the developed UPLC-MS/MS method. The results are presented in Table 2. The suitable incubation time and protein concentration for PH, CAN, BP, TBD, DEX, CLN, and TS were 15 min and 0.2 mg/mL, respectively. Also, the optimized probe concentration was 20 μM for PH, TBD, and TS and 10 μM for CAN, BP, DEX, and CLN. The K_m_ (Michaelis–Menten constant) values were 5.39 μM for PH, 0.41 μM for CAN, 9.15 μM for BP, 10.76 μM for TBD, 6.33 μM for DEX, 17.42 μM for CLN, and 85.21 μM for TS. For CAN, the K_m_ was 2.0–4.3 μM in dog liver microsomes, and the value was 0.41 μM in this study [48]. The K_m_ for CLN was 17.42 μM, which was smaller compared with 42–82 μM in dog liver microsomes, but it was similar to the 25–37 μM in mini-pig liver microsomes [48]. The K_m_ for TS was 192 μM in dog liver microsomes, but the value was similar to the rat and mini-pig values of 84 μM and 79 μM, respectively [48]. Our findings indicated that the V_max_ (maximum rate of an enzyme-catalyzed reaction) values were 0.22 nmol/min/mg protein for PH, 0.034 nmol/min/mg protein for CAN, 1.43 nmol/min/mg protein for BP, 293.1 nmol/min/mg protein for TBD, 6.16 nmol/min/mg protein for DEX, 5.4 nmol/min/mg protein for CLN, and 1.49 nmol/min/mg protein for TS. In contrast, the V_max_ values for CAN, CLN, and TS were 1488–1656 pmol/min/mg/microsomal protein, 573–1129 pmol/min/mg/microsomal protein, and 68–180 pmol/min/mg/microsomal protein in dog liver microsomes [48]. The K_m_ for PH, TBD, and DEX ranged from 10–50 μM, 100–200 μM, and 2–10 μM in in vitro drug metabolism studies [17]. The enzymatic kinetic results of probe substrates are presented in Figure 2. The incubation time and protein concentration are shown in Appendix A (these are shown in the Appendix A).

### 2.3. CYP450 Inhibition and Enzymatic Kinetic Study

Direct IC_50_ values only preliminarily indicate the CYP450 enzyme inhibitory potential in the xenobiotics. In the presence of LKMS, as shown in Figure 3 and Table 2, the activity of CYP2D6 was inhibited with an IC_50_ value of 39.66 μM. A similar inhibitory effect and concentration-dependent effect was observed in the activities of CYP2A6 (IC_50_ = 62.53 μM) and CYP2B6 (IC_50_ = 83.29 μM), while the IC_50_ values of CYP1A2, CYP2C9, CYP2E1, and CYP3A4 were >200 μM. LKMS slightly inhibited the activities of CYP2A6, CYP2B6, and CYP2D6, whereas it had no effect on the biotransformation activity of CYP1A2, CYP2C9, CYP2E1, or CYP3A4 in DLM. DDI-involved macrolides are attributed to mechanism-based CYP3A4, where macrolides inactivated HTS in liver microsomes [49]. Macrolides, such as troleandomycin, erythromycin, clarithromycin, roxithromycin, and azithromycin, are poor inhibitors of CYP1A2 when PH is used to measure its activity [50]. In line with this, our results indicated that the LKMS did not affect the activity of CYP1A2, and the LKMS showed no inhibition of CYP3A4 activity. Investigating the mechanism of inhibition of LKMS in CYP450 isoforms is useful to clarify the irreversible and reversible inhibition type involved in interactions between inhibitors and enzymes [51]. In addition, the inhibition mechanism was evaluated for CYP450 isoforms that presented IC_50_ values lower than 100 μM [52]. The enzyme’s kinetic analysis via Lineweaver–Burk plots and secondary plots was applied to fit the inhibition models of CYP2A6, CYP2B6, and CYP2D6. As shown in Figure 4 and Table 3, LKMS exhibited mixed-type inhibition against the activity of CYP2A6 with a K_i_ of 135.6 μΜ. LKMS inhibited CYP2B6 and CYP2D6 in a mixed way with K_i_ values of 59.44 μM and 64.87 μM, respectively. The mixed inhibition was evident such that the alpha inhibition constants (αK_i_) for CYP2A6, CYP2D6, and CYP2B6 were 227.6, 279.6, and 12.07 μM (K_i_ ≠ αK_i_), respectively [53]. In our study on inhibition mechanisms, we utilized duplicated samples owing to their good reproducibility and cost-effectiveness. However, to ensure the utmost accuracy of the results, we recommend employing triplicate samples. These data indicate that LKMS inhibits the activities of three major CYP450 enzymes through mixed inhibition, namely, CYP2A6, CYP2B6, and CYP2D6.

### 2.4. Irreversible Inhibition (IC_50_ Shift)

The IC_50_ shift value is widely used to determine time-dependent inhibitions (irreversible inhibition) [54]. According to our inhibition assay results, LKMS was a potent inhibitor of CYP450, which showed that LKMS relatively inhibited CYP2A6, CYP2B6, and CYP2D6 activities. Studying the inhibitory mechanism of the above three CYP450 isoforms was meaningful. As shown in Table 3 and Figure 5, the IC_50_ fold-shift (the ratio of the IC_50_ absence of NADPH divided by the IC_50_ presence of NADPH) of CYP2B6 was <1, indicating a right shift, while that of CYP2A6 and CYP2D6 was a left shift [55]. The IC_50_ fold-shift values were <1.5 [35], which indicated that LKMS might not be a time-dependent inhibitor of CYP2A6, CYP2B6, and CYP2D6.

### 2.5. Chemical Inhibition Experiments

An inhibition assay was performed on DLMs to confirm the contribution of the identified CYP450 enzymes in the metabolism of LKMS. The inhibitors used included α-Naphthoflavone (CYP1A2), pilocarpine (CYP2A6), thiotepa (CYP2B6), sulfaphenazole (CYP2C9), quinidine (CYP2D6), sodium diethyldithiocarbamate (CYP2E1), and ketoconazole (CYP3A4). The percentage inhibition of the LKMS disappearance rate in DLMs is presented in Figure 6. The co-incubation of LKMS with the CYP2D6 inhibitor decreased LKMS by 4.4%, while ketoconazole, a CYP3A4 inhibitor, inhibited LKMS metabolism by 17.4%. When incubated with inhibitors of the CYP1A2, CYP2C9, CYP2E1, CYP2B6, and CYP2A6 enzymes, the biotransformation rate of LKMS decreased by about 50%.

## 3. Discussion

Given the wide involvement of CYP450 in the metabolism processes of various drugs, the changes in the activity of hepatic metabolic enzymes may cause DDIs [56]. Accurately assessing the contribution of CYP450 isoforms is important for predicting clinical DDIs. The CYP450 enzymes involved in metabolism mainly include CYP3A4, CYP1A2, CYP2C9, CYP2C19, and CYP2D6, accounting for almost 78% of CYP450 enzymes [16,17]. The FDA and the European Medicines Agency have recommended assaying the DDI potential of an investigational drug candidate toward the key hepatic CYP450 enzymes involved in drug metabolism [19,57]. PH, CAN, BP, TBD, DEX, CLN, and TS are commonly employed as probe substrates to assess the activity of CYP1A2, CYP2A6, CYP2B6, CYP2C9, CYP2D6, CYP2E1, and CYP3A4 isoforms [58,59,60,61]. In this study, a cocktail approach was used to evaluate the influence of LKMS on the activity of CYP450 enzymes [36]. This is the first comprehensive report on the in vitro drug interaction potential of LKMS studied against seven CYP450 isoforms in DLMs.

Furthermore, the inhibitory effects of LKMS on CYP1A2, CYP2A6, CYP2B6, CYP2C9, CYP2D6, CYP2E1, and CYP3A4, herein the IC_50_ values of LKMS regarding CYP450 isoforms, were determined. The results indicated that LKMS is a potent inhibitor of CYP2A6, CYP2B6, and CYP2D6 with IC_50_ values of 62.53 μΜ, 83.29 μΜ, and 39.66 μΜ, respectively, but it did not exhibit the inhibition of CYP1A2, CYP2C9, CYP2E1, or CYP3A4 activities, as indicated by IC_50_ values exceeding 200 μΜ. Moreover, LKMS exhibited the mixed inhibition of CYP2A6, CYP2B6, and CYP2D6 activities. Of these CYP450 isoforms, LKMS showed the most potent inhibition of CYP2D6. As reported, the variation in CYP2D enzyme activity can increase the risk of adverse drug reactions [62]. The IC_50_ shift assay is one of the effective means of assessing the time-dependent inhibition of CYP450 enzymes [55]. IC_50_ fold-shifts of CYP2A6 and CYP2D6 were >1, while that of CYP2B6 was <1. Some drugs can be metabolized during preincubation to recover enzyme activity, resulting in an increased IC_50_ value [63,64]. Berry et al. mentioned that shifted IC_50_ values depend on the choice of inhibitor concentrations to the extent of reversible inhibition and incubation time [65]. Thus, the inhibition parameters of K_i_ are more important, particularly in applying extrapolation to in vivo DDIs [65]. In our study, LKMS demonstrated mixed-type inhibition against the activity CYP2A6 with a K_i_ value of 135.6 μΜ. Additionally, LKMS inhibited CYP2D6 and CYP2B6 in a mixed manner, with K_i_ values of 64.87 μM and 59.44 μM, respectively. Macrolide veterinary medicines are composed of lactone rings, which differ in their abilities to bind to and inhibit CYP3A4 [66]. Furthermore, most drug interactions regarding macrolides can inhibit CYP3A4-mediated catalytic activity [67,68]. Azithromycin has a weak effect on CYP3A4 and is less prone to drug interactions when combined with CYP3A4 enzyme-induced drugs [25,69]. Togami et al. used the probe drug midazolam to determine the activity of CYP3A4, which illustrates how azithromycin has a weak inhibitory effect on CYP3A4 [25]. Given that LKMS has a similar lactone ring structure to azithromycin, the preservation of this lactone ring structure in LKMS during metabolism might result in a mild inhibitory effect [25]. In addition, telithromycin can decrease the activities of hepatic CYP1A2 and CYP3A2 [10]. Similar to our results, the IC_50_ of erythromycin in CYP1A2, CYP2C9, CYP2C19, and CYP2D6 was greater than 300 μM, which indicates a weak inhibitory effect on CYP450 [70]. The phenotyping results indicated the main hepatic enzyme isoform was CYP2D6. The CYP2D6 isoform corresponds to only 2% of the CYP450 enzymes expressed in the liver, which are responsible for the metabolism of approximately 25% of medicines [71,72]. Therefore, considering that this study was conducted in vitro, it is advisable to undertake further in vivo investigations to elucidate the interaction between LKMS and canine CYP450 enzymes. Such preclinical biological screening plays a pivotal role in establishing the therapeutic effectiveness of new medicines and minimizing the potential for drug interactions.

## 4. Materials and Methods

### 4.1. Chemicals and Reagents

LKMS was provided by Henan Pulike Biological Engineering Co., Ltd. (Luoyang, China). LC-MS-grade methanol (MeOH), acetonitrile (ACN), and formic acid (FA) were purchased from Fisher Scientific (Pittsburgh, PA, USA). Ultrapure water was purified using a Milli-Q system (Millipore Corporation, Billerica, MA, USA). The following compounds were purchased from Sigma-Aldrich (St. Louis, MO, USA): PH, CAN, BP, TBD, DEX, CLN, and TS. ACE, HCAN, HBP, HTBD, HDEX, HCLN, and HTS were purchased from Beijing Huizhi Taikang Biological Technology (Beijing, China). α-Naphthoflavone, Pilocarpine, Ticlopidine, Sulfaphenazole, Quinidine, sodium diethyldithiocarbamate, and Ketoconazole were purchased from Sigma (St. Louis, MO, USA). Phosphate buffer (PBS) tablets were purchased from Beijing Solarbio Technology (Beijing, China). β-nicotinamide adenine dinucleotide phosphate was purchased from Fisher Scientific (Pittsburgh, PA, USA).

PH, CAN, BP, TBD, DEX, CLN, and TS stock solutions were prepared in ACN at 10 mg/mL. Stock solutions of ACE, HCAN, HBP, HTBD, HDEX, HCLN, and HTS were prepared in MeOH at a concentration of 10 μg/mL. The internal standard of aflatoxin B1 was prepared at 200 ng/mL in ACN. The specific probe substrates, metabolites, and inhibitors are presented in Table 4.

### 4.2. Instruments and Chromatographic Conditions

Quantitative analysis was carried out using UPLC coupled with TSQ Quantis™ (Thermo Fisher, Waltham, MA, USA). A HSS T3 column (100 mm × 2.1 mm, 1.7 μm) (Waters, USA) was used at 35 °C. Both positive and negative electrospray ionization (ESI+ and ESI-) modes were used in this study. The other parameters were as follows: capillary voltage, 3.5 kV (ESI+) and 2.7kV (ESI−); ion source temperature, 320 °C; vaporizer temperature, 350 °C; sheath gas flow rate, 50 arb; and collision gas flow rate, 12 arb. An injection volume of 5 µL was applied. Analytes were eluted using a mobile phase of 0.1% FA·H_2_O (A): ACN (B) (90:10, *v*/*v*) at a flow rate of 0.2 mL/min. The gradient was as follows: 0–1.0 min (10%, B), 1.0–3.0 min (10–90%, B), 3.0–4.0 min (90%, B), 4.0–4.5 min (90–10%, B), 4.6–6.0 min (10%, B). The mass spectrometry parameters are provided in Table 5.

### 4.3. Microsomal Incubation and Enzymatic Kinetic Study

The CYP450 enzymatic activities were characterized by their probe substrate reactions. Incubation mixtures were prepared in a total volume of 200 μL, which consisted of PBS (pH = 7.4), NADPH, probe substrates, and various concentrations of tested compounds. NADPH was added to initiate the reaction after a 5 min preincubation period at 37 °C in a thermos shaker. Of note, the volume of the final organic solvent in the incubation system should be below <1% (*v*/*v*) [21]. All experiments were performed in triplicate.

The probe substrate concentrations were as follows: PH, BP, CLN, and CAN, 1–200 μM; TBD, 10–400 μM; DEX, 1–100 μM; and TS, 5–400 μM. The V_max_ and K_m_ were calculated using GraphPad Prism v.8.0.2 (GraphPad software Inc., San Diego, CA, USA) via the Michaelis–Menten kinetic model. In addition, incubation times of 0–60 min were examined for mixed probe substrates. Various DLM protein concentrations (0.05–1 mg/mL) in the incubation mixture were carried out.

Then, 400 μL of cool ACN containing the 20 μL internal standard was added to terminate the reaction. Samples were vortexed for 2 min and centrifuged at 14,000 rpm for 10 min (4 °C). Next, the supernatant was steamed under nitrogen (40 °C) until dry. Then, it was dissolved with 200 µL of ACN: H_2_O = 80:20 (*v*/*v*). Finally, after filtering with a 0.22 μm membrane, an aliquot of 5 µL was injected into the UPLC-MS/MS for analysis.

### 4.4. CYP450 Inhibition Study

Inhibition assessment of CYP450 isoforms in DLM was performed with a fixed concentration of model probe substrates and a series of concentrations of LKMS ranging from 0 to 80 μM. The concentration of probe substrates was at a concentration of K_m_ in this study. Meanwhile, the K_i_ values were measured with various concentrations of probe substrates (0.3 K_m_, 1 K_m_, 3 K_m_, 6 K_m_, and 10 K_m_) and LKMS (0, 0.25 IC_50_, 0.5 IC_50_, 1 IC_50_, and 2.5 IC_50_). K_i_ values of LKMS were determined if their IC_50_ values were lower than 100 μM [52,73]. The IC_50_ was calculated using the GraphPad Prism v.8.0.2 (GraphPad Software Inc., San Diego, CA, USA). Mode inhibition was examined based on a primary Lineweaver–Burk plot and a secondary plot.

### 4.5. Irreversible Inhibition (IC_50_ Shift)

To characterize the reversible or irreversible mechanism of CYP450 inhibition via LKMS, the shifted IC_50_ values were used to determine whether the inhibition of LKMS on CYP450 was time-dependent. The IC_50_ shift experiment was performed via the preincubation of LKMS in DLMs for 30 min at 37 °C with and without NADPH. The above mixture was then incubated for another 30 min in the presence of NADPH. The concentrations of LKMS were designated as 0.2, 2, 4, 6, 8, 10, 20, 40, and 80 μM, and the concentration of each probe substrate was based on each K_m_ value. The concentrations of the probe substrates of CYP2A6, CYP2B6, and CYP2D6 were at single concentrations approximating their K_m_ values [55].

### 4.6. Chemical Inhibition Experiments

The effect of selective inhibitors on the rate of LKMS disappearance was investigated to identify the metabolizing enzymes of LKMS. The total volume of the incubation system was 200 μL, which contained the tested compound, inhibitors, PBS (pH = 7.4), LKMS, and NADPH. The study was performed via incubation with different inhibitors of CYP450 as follows: α-naphthoflavone (CYP1A2, 2 μM), pilocarpine (CYP2A6, 8 μM), Thio-TEPA (CYP2B6, 6 μM), sulfaphenazole (CYP2C9, 6 μM), quinidine (CYP2D6, 8 μM), sodium diethyldithiocarbamate (CYP2E1, 20 μM), and ketoconazole (CYP3A4, 2 μM). The sample incubated without inhibitors was used as a negative control. The sample was incubated with dimethyl sulfoxide (DMSO) (25%) as a positive control in this study. Following this pilot, confirmatory experiments were conducted by incubating with DLM in the presence of specific CYP450 inhibitors.

### 4.7. Statistical Analysis

The IC_50_ values and enzyme kinetic parameters were determined by plotting relative activities over the logarithm of the concentration of LKMS using GraphPad (v.8.0.2, CA, USA). The mode of the inhibitor was analyzed using a Lineweaver–Burk plot and a secondary plot for inhibition constant (K_i_ and αK_i_) values. All experimental data are shown as mean ± standard deviation (SD).

## 5. Conclusions

In the current study, a UPLC-MS/MS method was established to simultaneously determine the metabolites of CYP450 probe substrates and to explore the effect of LKMS on the activities of dog CYP450 enzymes, both in vitro and in vivo. Importantly, the mechanisms of the inhibition of CYP450 caused by LKMS were investigated for the first time. Accordingly, our data suggest that LKMS can act as a mixed inhibitor, inhibiting CYP2A6, CYP2B6, and CYP2D6. Moreover, this inhibition via LKMS did not exhibit time-dependent behavior. All these results indicated the potential for DDIs between LKMS and drugs metabolized by CYP2A6, CYP2B6, or CYP2D6. However, further in vivo studies are needed to verify these potential effects.

## Figures and Tables

**Figure 1 molecules-28-07193-f001:**
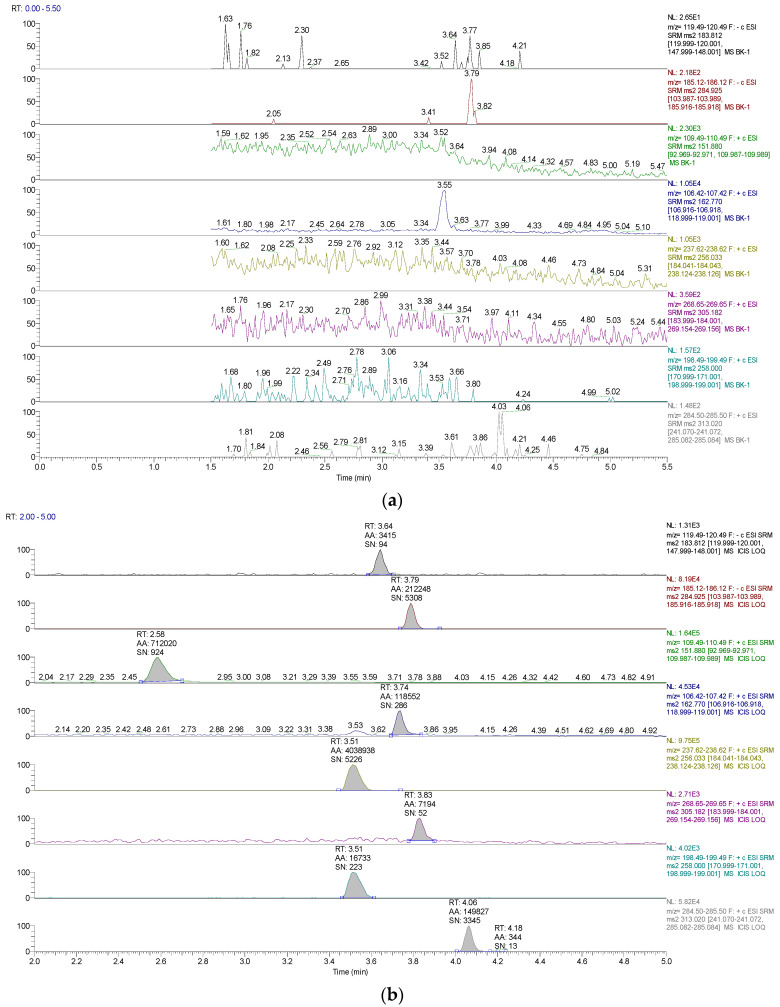
Typical chromatograms of blank (**a**) and spiked metabolites of probe substrates of CYP450 enzymes (**b**) in dog liver microsomes.

**Figure 2 molecules-28-07193-f002:**
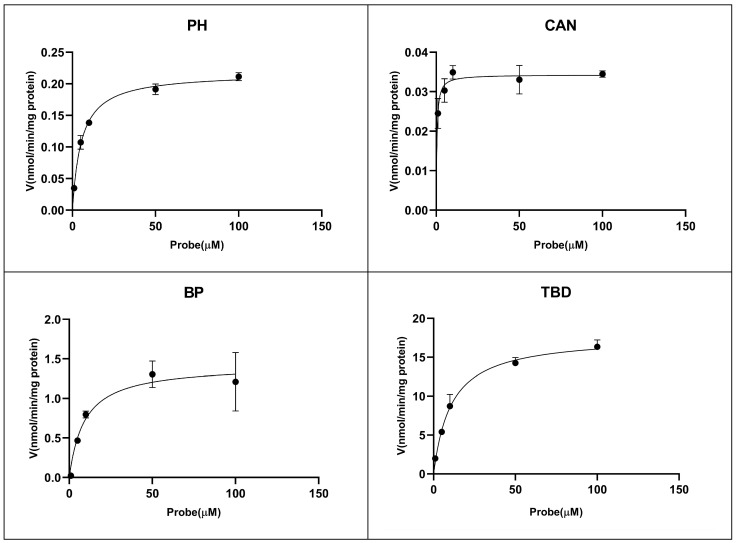
Enzymatic kinetics of phenacetin (PH) for CYP1A2, coumarin (CAN) for CYP2A6, bupropion (BP) for CYP2B6, tolbutamide (TBD) for CYP2C9, dextromethorphan (DEX) for CYP2D6, chlorzoxazone (CLN) for CYP2E1, and testosterone (TS) for CYP3A4 in dog liver microsomes. Data are presented as mean ± standard deviation (SD) (n = 3).

**Figure 3 molecules-28-07193-f003:**
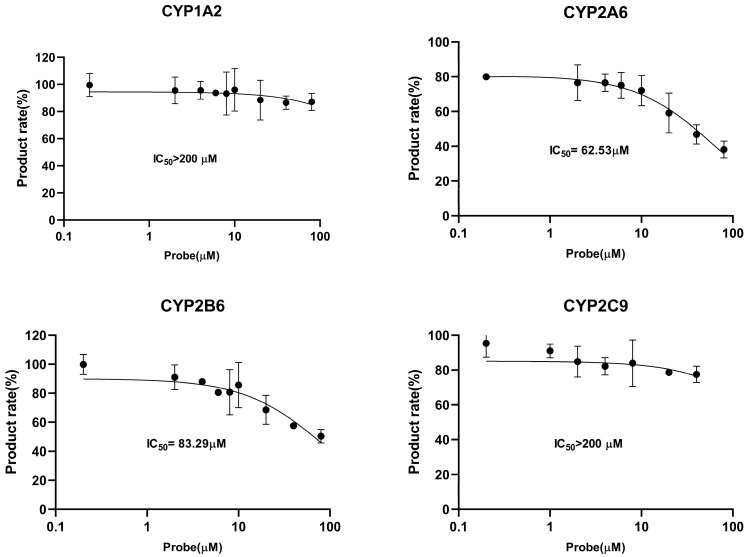
Inhibition curves of lekethromycin for canine CYP450 activities using probe substrates in dog liver microsomes, phenacetin-O-deethylation for CYP1A2, 5-hydroxylation coumarin for CYP2A6, 1-hydroxylation bupropion for CYP2B6, 4-hydroxylation tolbutamide for CYP2C9, dextromethorphan O-demethylation for CYP2D6, 1-hydroxylation chlorzoxazone for CYP2E1, and 6β-hydroxylation testosterone for CYP3A4. The data are plotted as the mean ± standard deviation (SD) (n = 3).

**Figure 4 molecules-28-07193-f004:**
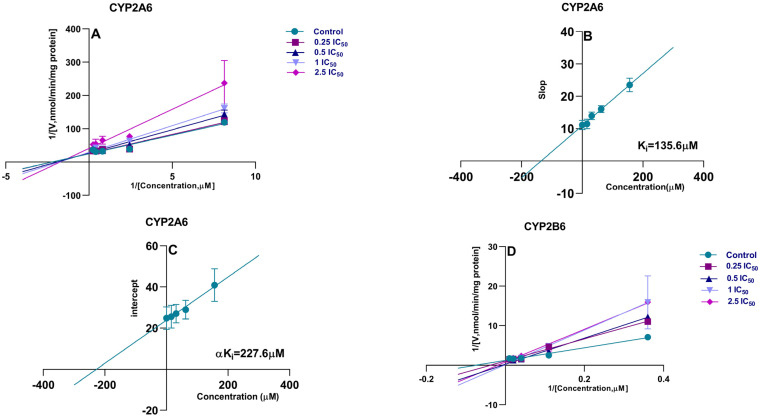
Primary Lineweaver−Burk plot, the secondary plot for Ki and αKi in dog liver mi-crosomes, Lineweaver−Burk plot for CYP2A6 (**A**), Slop plot for CYP2A6 (**B**), intercept plot for CYP2A6 (**C**), Lineweaver−Burk plot for CYP2B6 (**D**), Slop plot for CYP2B6 (**E**), intercept plot for CYP2B6 (**F**), Lineweaver−Burk plot for CYP2D6 (**G**), Slop plot for CYP2D6 (**H**), intercept plot for CYP2D6 (**I**). Each point is represented as mean ± standard deviation (SD) (n = 2).

**Figure 5 molecules-28-07193-f005:**
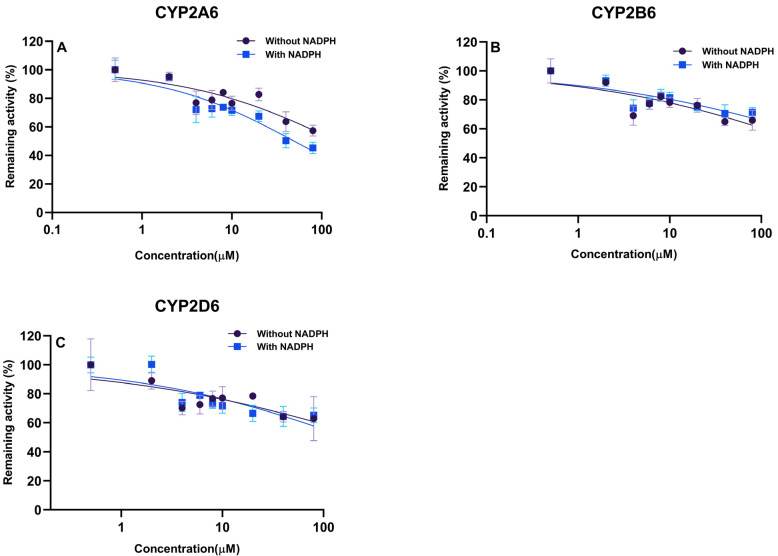
The IC_50_ shift curve of lekethromycin in dog liver microsomes, 5-hydroxylation coumarin for CYP2A6 (**A**), 1-hydroxylation bupropion for CYP2B6 (**B**), dextromethorphan O-demethylation for CYP2D6 (**C**). Each point represents the mean ± standard deviation (SD) (n = 3).

**Figure 6 molecules-28-07193-f006:**
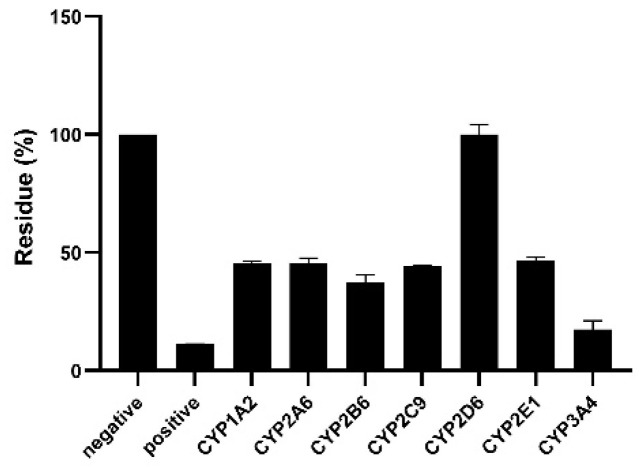
The contribution of dog CYP450 to the metabolism of lekethromycin. Each point represents the mean ± standard deviation (SD) (n = 3).

**Table 1 molecules-28-07193-t001:** The accuracy and precision of specific metabolites of CYP450 in dog liver microsomes.

Analyte	Concentration (ng/mL)	Recovery (%)	Intra-Day RSD (%)	Inter-Day RSD (%)
ACE	10	104.80 ± 6.36	6.07	5.73
	50	101.66 ± 2.00	1.96	4.19
	200	101.92 ± 1.44	1.41	2.69
HCAN	10	99.86 ± 8.06	8.07	7.29
	50	99.39 ± 7.04	7.09	7.24
	200	99.90 ± 6.47	6.48	4.93
HBP	10	104.86 ± 8.55	8.16	7.84
	50	103.64 ± 3.43	3.31	5.56
	200	101.29 ± 3.18	3.14	3.82
HTBD	10	105.04 ± 11.72	11.15	10.04
	50	102.01 ± 9.83	9.64	8.48
	200	95.53 ± 1.30	1.36	2.04
HDEX	10	109.42 ± 1.8	12.14	10.84
	50	106.93 ± 4.05	1.68	6.67
	200	97.79 ± 3.71	3.79	6.34
HCLN	10	97.35 ± 8.65	8.88	9.31
	50	101.99 ± 11.41	11.18	9.81
	200	97.63 ± 1.48	1.52	7.14
HTS	10	95.63 ± 6.89	7.20	9.08
	50	97.40 ± 1.76	1.80	4.59
	200	101.89 ± 2.45	2.40	3.96

RSD: relative standard deviation; acetaminophen (ACE); 7-OH-coumarine (HCAN); OH-bupropion (HBP); 4-OH-tolbutamide (HTBD); dextrorphan (HDEX); 6-OH-chlorzoxazone (HCLN); 6β-OH-testosterone (HTS).

**Table 2 molecules-28-07193-t002:** CYP450 isoforms, reactions, incubation conditions, and inhibition parameters in dog liver microsomes (n = 3).

Isoforms	Reactions	Time (min)	Protein Concentration (mg/mL)	K_m_ (μM)	V_max_ (nmol/min/mg protein)	IC_50_ (μM)
CYP1A2	Phenacetin- O-deethylation	15	0.2	5.39	0.22	>200
CYP2A6	Coumarin 5-hydroxylation	15	0.2	0.41	0.034	62.53
CYP2B6	Bupropion 1-hydroxylation	15	0.2	9.15	1.43	83.29
CYP2C9	Tolbutamide 4-hydroxylation	15	0.2	10.76	17.78	>200
CYP2D6	Dextromethorphan O-demethylation	15	0.2	6.33	6.16	39.66
CYP2E1	Chlorzoxazone 1-hydroxylation	15	0.2	17.42	5.4	>200
CYP3A4	Testosterone 6β-hydroxylation	15	0.2	85.21	1.49	>200

V_max_: maximum rate of an enzyme-catalyzed reaction; K_m:_ Michaelis–Menten constant; IC_50_: half-maximal inhibitory concentration.

**Table 3 molecules-28-07193-t003:** The IC_50_ fold-shift and inhibition parameters in CYP450 enzymes in dog liver microsomes.

CYP450 Isoform	Shift Ratio (−NADPH/+NADPH)	K_i_ (μM)	αK_i_ (μM)
CYP2A6	1.29	135.6	227.6
CYP2D6	1.10	64.87	279.6
CYP2B6	0.88	59.44	12.07

K_i_: inhibition constant; αK_i_: alpha inhibition constant.

**Table 4 molecules-28-07193-t004:** List of specific CYP450 probe substrates, metabolites, and inhibitors.

CYP450 Isoform	Substrates	Metabolites	Inhibitors
CYP1A2	PH	ACE	α-Naphthoflavone
CYP2A6	CAN	HCAN	Pilocarpine
CYP2B6	BP	HBP	Thio-TEPA
CYP2C9	TBD	HTBD	Sulfaphenazole
CYP2D6	DEX	HDEX	Quinidine
CYP2E1	CLN	HCLN	Sodium diethyldithiocarbamate
CYP3A4	TS	HTS	Ketoconazole

Phenacetin (PH); coumarin (CAN); bupropion (BP); Tolbutamide (TBD); dextromethorphan (DEX); chlorzoxazone (CLN); and testosterone (TS). Acetaminophen (ACE); 7-OH-coumarine (HCAN); OH-bupropion (HBP); 4-OH-tolbutamide (HTBD); dextrorphan (HDEX); 6-OH-chlorzoxazone (HCLN); and 6β-OH-testosterone (HTS).

**Table 5 molecules-28-07193-t005:** Mass spectrometry parameters for all analytes.

Substrates	Quantitative Ion	Collision Energy (eV)	Ion Source
PH	180 > 110	20	ESI+
CAN	147 > 110	19	ESI+
BP	240 > 184	18	ESI+
TBD	271 > 91	31	ESI+
DEX	272 > 215	23.5	ESI+
CLN	168 > 132	20	ESI−
TS	289 > 97	22	ESI+
ACE	152 > 110	16	ESI+
HCAN	163 > 119	34	ESI+
HBP	256 > 184	15	ESI+
HTBD	285 > 186	17	ESI−
HDEX	258 > 171	30	ESI+
HCLN	184 > 120	15	ESI−
HTS	305 > 184	10	ESI+

Phenacetin (PH); coumarin (CAN); bupropion (BP); Tolbutamide (TBD); dextromethorphan (DEX); chlorzoxazone (CLN); and testosterone (TS). Acetaminophen (ACE); 7-OH-coumarine (HCAN); OH-bupropion (HBP); 4-OH-tolbutamide (HTBD); dextrorphan (HDEX); 6-OH-chlorzoxazone (HCLN); and 6β-OH-testosterone (HTS).

## Data Availability

Not applicable.

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
