# Peer review of "Inhibitory Mechanisms of Lekethromycin in Dog Liver Cytochrome P450 Enzymes Based on UPLC-MS/MS Cocktail Method"

_molecules, 2023, doi:10.3390/molecules28207193_

Round 1
Reviewer 1 Report
Comments and Suggestions for Authors
This manuscript evaluates the inhibitory effect of Lekethromycin on dog CYP450 enzymes. Although a variety of typical probe substrates were selected for the assay, the detailed inhibitory effect and mechanism of Lekethromycin on CYP450 was not explained clearly and requires further elaboration.
§ In the Introduction section, more references need to be added to introduce the importance of the work as well as the innovations.
§ The description in line 48 is incorrect; not all CYP450 expression is concentrated in the liver or intestine.
§ In line 66, the abbreviation “DDI” has not been defined. Furthermore, please carefully review and define other abbreviations used throughout the manuscript.
§ To improve the readability of the data, the images in Figure 1 need to be enlarged.
§ In Figure 2, there may be an issue with the data for TBD; please verify this.
§ The types of CYP450 mentioned in Section 2.3 have not been explained in Figure 3.
§ Further clarification is needed regarding the mechanism by which Lekethromycin inhibits CYP450.
§ The section 2 should not merely provide a simple description of the data; it should include comparisons with other literature.
§ I think the conclusion section needs to be revised to better summarise the significance of this work.
Author Response
Cover letter
Dear editors and reviewers,
Thanks for providing us with this great opportunity to submit a revised version of our manuscript. We appreciate the detailed and constructive comments provided by the reviewers. We have carefully revised the manuscript by incorporating all the suggestions by the review panel.
We hope this revised manuscript has addressed your concerns, and look forward to hearing from you.
Sincerely,
The authors
Reviewer 1
Comments and Suggestions for Authors
This manuscript evaluates the inhibitory effect of Lekethromycin on dog CYP450 enzymes. Although a variety of typical probe substrates were selected for the assay, the detailed inhibitory effect and mechanism of Lekethromycin on CYP450 was not explained clearly and requires further elaboration.
Comment 1:
In the Introduction section, more references need to be added to introduce the importance of the work as well as the innovations.
Response 1:
- P. Sun et al., “Plasma Protein Binding Rate and Pharmacokinetics of Lekethromycin in Rats,” Antibiotics, vol. 11, no. 9, p. 1241, Sep. 2022, doi: 10.3390/antibiotics11091241.
- Y. Cao et al., “Determination of lekethromycin in plasma and tissues of pneumonia-infected rats by ultra-high performance liquid chromatography-tandem mass spectrometry,” Journal of Chromatography B, vol. 1227, p. 123811, Jul. 2023, doi: 10.1016/j.jchromb.2023.123811.
- Tang, D.Q.; Li, Y.J.; Li, Z.; Bian, T.T.; Chen, K.; Zheng, X.X.; Yu, Y.; Jiang, S.S. Study on the interaction of plasma pro-tein binding rate between edaravone and taurine in human plasma based on HPLC analysis coupled with ultrafil-tration technique. Biomed. Chromatogr. 2015, 29, 1137–1145
- Sane, R.S.; Ramsden, D.; Sabo, J.P.; Cooper, C.; Rowland, L.; Ting, N.; Whitcher-Johnstone, A.; Tweedie, D.J. Contri-bution of Major Metabolites toward Complex Drug-Drug Interactions of Deleobuvir: In Vitro Predictions and In Vi-vo Outcomes. Drug Metab. Dispos. 2015, 44, 466–475.
- Wang, Y.; Wang, C.; Wang, S.; Zhou, Q.; Dai, D.; Shi, J.; Xu, X.; Luo, Q. Cytochrome P450-Based Drug-Drug Interac-tions of Vonoprazan In Vitro and In Vivo. Front. Pharmacol. 2020, 11, 53.
Comment 2:
The description in line 48 is incorrect; not all CYP450 expression is concentrated in the liver or intestine.
Response 2:
Thanks for your useful advice. The sentence was revised as below: The liver, intestine, lung, kidney, brain, heart, adrenal, gonads, and most others have the capability for biotransformation through CYP450 enzyme [9]–[12]. Among these, liver is the major organ involved in CYP450-mediated metabolism.
Comment 3:
In line 66, the abbreviation “DDI” has not been defined. Furthermore, please carefully review and define other abbreviations used throughout the manuscript.
Response 3:
Thanks for your useful advice. The error has been rectified. Additionally, the abbreviations used in this paper were carefully reviewed and enhanced for clarity.
Comment 4:
To improve the readability of the data, the images in Figure 1 need to be enlarged.
Response 4:
Thanks for your useful advice. The resolution of Figure 1 has been enhanced, and the specifications have been adjusted as the journal's requirements.
Comment 5:
In Figure 2, there may be an issue with the data for TBD; please verify this.
Response 5:
Thanks for your useful advice. I apologize for my mistake, and I have thoroughly reviewed and revised the results. Initially, I employed the area to calculate the enzymatic kinetic parameters due to the concentration of HTBD being below the LOQ (10 ng/mL). However, I have since repeated the experiment and recalculated the Km and Vmax values for TBD. The data will be made if necessary.
Comment 6:
The types of CYP450 mentioned in Section 2.3 have not been explained in Figure 3.
Response 6:
Thanks for your useful advice. The CYP450 isoforms and IC50 values have been incorporated into Figure 3. The caption has been revised as below:
Figure 3. Inhibition curves of lekethromycin for canine CYP450 activities using probe substrates in dog liver microsomes, 1-hydroxylation bupropion for CYP2B6, 5-hydroxylation coumarin for CYP2A6, 1-hydroxylation chlorzoxazone for CYP2E1, dextromethorpham O-demethylation for CYP2D6, phenacetin-O-deethylation for CYP1A2, 4-hydroxylation tolbutamide for CYP2C9, and 6β-hydroxylation testosterone for CYP3A4. The data plotted as the mean ± SD (n=3).
Comment 7:
Further clarification is needed regarding the mechanism by which Lekethromycin inhibits CYP450.
Response 7:
Thanks for your useful advice. I apologize for my oversight in not including the complete Lineweaver-Burk plot and the secondary plot. I have now rectified this by incorporating the results pertaining to the LKMS mechanism, which can be observed in Figure 4.
Figure 4. Primary Lineweaver-Burk plot, the secondary plot for Ki and αKi in dog liver microsomes. Each point is represented as mean ± standard deviation (SD) (n=2).
Comment 8:
The section 2 should not merely provide a simple description of the data; it should include comparisons with other literature.
Response 8:
Thanks for your helpful and useful advice. I have incorporated a discussion section into the results to enable a comparison with the existing literature, for instance, this includes a comparison of the kinetic parameters of probe substrates for CYP450 enzymes with those from other studies.
Comment 9:
I think the conclusion section needs to be revised to better summarise the significance of this work.
Response 9:
Thanks for your helpful advice. The conclusion was rewritten to highlight the significance of this work as below:
In the current study, an UPLC-MS/MS method was established to simultaneously determine metabolites of CYP450 probe substrates, and to explore the effect of LKMS on the activities of dog CYP450 enzymes, both in vitro and in vivo studies. Importantly, the mechanisms of inhibition of CYP450 by LKMS were investigated for the first time, and our data suggested that LKMS act as a mixed inhibitor, inhibiting CYP2A6, CYP2B6, and CYP2D6. Moreover, this inhibition by LKMS don’t exhibit a time-dependent manner. All these results indicate that potential for DDI between LKMS and drugs metabolized by CYP2A6, CYP2B6, or CYP2D6. In conclusion, further in vivo studies are needed to verify these potential effects.

Reviewer 2 Report
Comments and Suggestions for Authors
The authors study the inhibitory mechanisms of lekethromycin (LKMS) on dog liver Cytochrome P450 enzymes using UPLC-MS/MS. This manuscript holds potential interest for Molecules' readers. However, a revision addressing the below points will enhance the manuscript's clarity and accuracy, making it more accessible to readers.
1. More introduction of LKMS is needed as well as the signification of determining its inhibitory mechanism.
2. Abbreviations should be defined at first mention. Please revise the manuscript carefully the manuscript containing many substances and the current wording make it difficult to identify them. Full synonyms needed: “DDI” (line 60); “IS” (line 283)
3. Please check the mention of Table in the text. Line 98 should be Table 1; Line 107 should Table 2; Line 135 should be Table 3; Line 150 should be Table 3; Line 254 should be Table 4; Line 266 should be Table 5.
4. Section 2.3, the text discusses the inhibition of CYP450 isoforms. However, Figure 3, which pertains to this section, is based on substrate, further complicating comprehension. It is the same for Section 2.4 and Figure 5.
5. The manuscript heavily relies on Table 4 to establish the associations between CYP450 isoforms and their respective substrates and inhibitors. Nevertheless, in another experiments (Section 2.5 and 4.6), a distinct inhibitor for CYP2A6 was identified, and a new isoform, CYP2C19, was also observed. Please ensure the accuracy of these writing.
6. Figure 4 and 5, n=2 is not sufficient to provide meaningful conclusion.
7. Section 4.2, the gradient of the mobile phase is a little bit oddly: “…4.0-4.5 min (90%-10%, B), 4.5-6.0 min (90%-10%, B).” (line265). Please check.
Comments on the Quality of English Language
Minor editing of English language required
Author Response
Cover letter
Dear editors and reviewers,
Thanks for providing us with this great opportunity to submit a revised version of our manuscript. We appreciate the detailed and constructive comments provided by the reviewers. We have carefully revised the manuscript by incorporating all the suggestions by the review panel.
We hope this revised manuscript has addressed your concerns, and look forward to hearing from you.
Sincerely,
The authors
Reviewer 2
comments and Suggestions for Authors
The authors study the inhibitory mechanisms of lekethromycin (LKMS) on dog liver Cytochrome P450 enzymes using UPLC-MS/MS. This manuscript holds potential interest for Molecules' readers. However, a revision addressing the below points will enhance the manuscript's clarity and accuracy, making it more accessible to readers.
Comment 1:
More introduction of LKMS is needed as well as the signification of determining its inhibitory mechanism.
Response 1: Thanks for your useful suggestion. References have been included to provide more introduction to this medication and to underscore the significance of elucidating its inhibition mechanism. The references were list as below:
- P. Sun et al., “Plasma Protein Binding Rate and Pharmacokinetics of Lekethromycin in Rats,” Antibiotics, vol. 11, no. 9, p. 1241, Sep. 2022, doi: 10.3390/antibiotics11091241.
- Y. Cao et al., “Determination of lekethromycin in plasma and tissues of pneumonia-infected rats by ultra-high performance liquid chromatography-tandem mass spectrometry,” Journal of Chromatography B, vol. 1227, p. 123811, Jul. 2023, doi: 10.1016/j.jchromb.2023.123811.
- Tang, D.Q.; Li, Y.J.; Li, Z.; Bian, T.T.; Chen, K.; Zheng, X.X.; Yu, Y.; Jiang, S.S. Study on the interaction of plasma pro-tein binding rate between edaravone and taurine in human plasma based on HPLC analysis coupled with ultrafil-tration technique. Biomed. Chromatogr. 2015, 29, 1137–1145
- Sane, R.S.; Ramsden, D.; Sabo, J.P.; Cooper, C.; Rowland, L.; Ting, N.; Whitcher-Johnstone, A.; Tweedie, D.J. Contri-bution of Major Metabolites toward Complex Drug-Drug Interactions of Deleobuvir: In Vitro Predictions and In Vi-vo Outcomes. Drug Metab. Dispos. 2015, 44, 466–475.
- Wang, Y.; Wang, C.; Wang, S.; Zhou, Q.; Dai, D.; Shi, J.; Xu, X.; Luo, Q. Cytochrome P450-Based Drug-Drug Interac-tions of Vonoprazan In Vitro and In Vivo. Front. Pharmacol. 2020, 11, 53.
Comment 2:
Abbreviations should be defined at first mention. Please revise the manuscript carefully the manuscript containing many substances and the current wording make it difficult to identify them. Full synonyms needed: “DDI” (line 60); “IS” (line 283)
Response 2:
Thanks for your useful suggestion. The paper was carefully reviewed and additional abbreviations utilized throughout the manuscript have been clearly defined.
Comment 3:
Please check the mention of Table in the text. Line 98 should be Table 1; Line 107 should Table 2; Line 135 should be Table 3; Line 150 should be Table 3; Line 254 should be Table 4; Line 266 should be Table 5.
Response 3:
Thanks for your useful suggestion. I apologize for my mistake. The entire text has undergone a thorough proofreading, and the issue has been rectified.
Comment 4:
Section 2.3, the text discusses the inhibition of CYP450 isoforms. However, Figure 3, which pertains to this section, is based on substrate, further complicating comprehension. It is the same for Section 2.4 and Figure 5.
Response 4:
Thanks for your useful suggestion. The Figure 3 and Figure 5 have been revised and replaced to include the CYP450 isoform instead of the probe substrates, aiming to eliminate any potential confusion. The Figure 3 and Figure 5 were shown as below:
Figure 3. Inhibition curves of lekethromycin for canine CYP450 activities using probe substrates in dog liver microsomes, phenacetin-O-deethylation for CYP1A2, 5-hydroxylation coumarin for CYP2A6, 1-hydroxylation bupropion for CYP2B6, 4-hydroxylation tolbutamide for CYP2C9, dextromethorphan O-demethylation for CYP2D6, 1-hydroxylation chlorzoxazone for CYP2E1, and 6β-hydroxylation testosterone for CYP3A4. The data was plotted as the mean ± standard deviation (SD) (n=3).
Figure 5. The IC50 shift curve of lekethromycin in dog liver microsomes, 5-hydroxylation coumarin for CYP2A6, 1-hydroxylation bupropion for CYP2B6, dextromethorpham O-demethylation for CYP2D6. Each point represents the mean±SD (n=3).
Comment 5:
The manuscript heavily relies on Table 4 to establish the associations between CYP450 isoforms and their respective substrates and inhibitors. Nevertheless, in another experiments (Section 2.5 and 4.6), a distinct inhibitor for CYP2A6 was identified, and a new isoform, CYP2C19, was also observed. Please ensure the accuracy of these writing.
Response 5:
Thanks for your useful and helpful suggestion. There were some errors during the copy and paste process, and I apologize for any oversight. The inhibitor used in the study for CYP2A6 is pilocarpine, and this information has been updated in the manuscript. Additionally, the content related to CYP2C19 has been removed from the paper.
Comment 6:
Figure 4 and 5, n=2 is not sufficient to provide meaningful conclusion.
Response 6:
Thanks for your useful suggestion. Regarding Figure 4, the protocol employed was as follows: https://www.cyprotex.com/admepk/in-vitro-metabolism/cytochrome-p450-ki, where the data has been represented in the graph as the mean ± SD of duplicate determinations. Additionally, we discussed and indicated the need for further experiments with three samples as below: In the study on inhibition mechanisms, we utilized duplicated samples owing to their good reproducibility and cost-effectiveness. However, to ensure the utmost accuracy of the results, it is recommended to employ triplicate samples.
Concerning Figure 5, the data has been presented as the mean ± SD of triplicate determinations. I apologize for any inadvertent errors that may have occurred during the copy and paste process.
Comment 7:
Section 4.2, the gradient of the mobile phase is a little bit oddly: “…4.0-4.5 min (90%-10%, B), 4.5-6.0 min (90%-10%, B).” (line265). Please check.
Response 7:
Thanks for your helpful suggestions. The change made is from "4.5-6.0 min (90%-10%, B)" to "4.6-6.0 min (10%, B)."
Comment 8:
Comments on the Quality of English Language: Minor editing of English language required
Response 8:
Thanks for your useful advice. The article has undergone proofreading and polishing by a native English speaker.

Reviewer 3 Report
Comments and Suggestions for Authors
While the search for new pharmacologically active substances, specifically potential antibiotics, is important topic, the reviewed manuscript is in too rough state.
While the meaning of the manuscript is somewhat understandable, the language requires significant improvement, stylistically and grammatically. This, combined with the state of Tables and Figures, makes it hard to properly access the validity of presented data.
Tables are not properly referenced in the manuscript – Table 4 and 5 are mentioned in text before Tables 1, 2 and 3.
Table 1 is referenced in line 254 but mentioned data is presented in Table 4, Table 2 is referenced in line 266 but mentioned data is presented in Table 5. Table 3 is not mentioned anywhere in the text. Table 4 is referenced in line 107 as containing results of enzyme kinetics but does not contain this data. Table 5 is referenced in line 150 as containing results of IC50 fold-shift but does not contain this data.
Line 123 states that Figure 3 contains data on CYP450 isoforms inhibition but no such data shown.
Figure 4 requires improvement – is almost impossible to discern which plot corresponds to which legend line.
In the beginning of Abstract – “Lekethromycin (LKMS) is a derivative synthetic macrolide compound with an excellent 13 pharmacological property of antibacterial.” – no reference is provided.
Introduction, line 38 – “And the result was observed in dogs in the further study in our lab.” – no reference is provided.
List of abbreviation is absent and abbreviations should be described on first mention in text.
Given the new compound, ADME (absorption, distribution, metabolism, excretion) analysis was necessary to evaluate the safety and risk when exposed to dog.
Comments on the Quality of English Language
While the meaning of the manuscript is somewhat understandable, the language requires significant improvement, stylistically and grammatically. This, combined with the state of Tables and Figures, makes it hard to properly access the validity of presented data.
Author Response
Cover letter
Dear editors and reviewers,
Thanks for providing us with this great opportunity to submit a revised version of our manuscript. We appreciate the detailed and constructive comments provided by the reviewers. We have carefully revised the manuscript by incorporating all the suggestions by the review panel.
We hope this revised manuscript has addressed your concerns, and look forward to hearing from you.
Sincerely,
The authors
Reviewer 3
Comments and Suggestions for Authors
While the search for new pharmacologically active substances, specifically potential antibiotics, is important topic, the reviewed manuscript is in too rough state.
While the meaning of the manuscript is somewhat understandable, the language requires significant improvement, stylistically and grammatically. This, combined with the state of Tables and Figures, makes it hard to properly access the validity of presented data.
Comment 1:
Tables are not properly referenced in the manuscript – Table 4 and 5 are mentioned in text before Tables 1, 2 and 3.
Response 1:
Thanks for your useful advice. The position and order of the tables and figures have been meticulously reviewed and adjusted in the revised draft.
Comment 2:
Table 1 is referenced in line 254 but mentioned data is presented in Table 4, Table 2 is referenced in line 266 but mentioned data is presented in Table 5. Table 3 is not mentioned anywhere in the text. Table 4 is referenced in line 107 as containing results of enzyme kinetics but does not contain this data. Table 5 is referenced in line 150 as containing results of IC50 fold-shift but does not contain this data.
Response 2:
Thanks for your useful advice. I apologize for my carelessness. The position and order of the table and figures have been carefully checked and modified in the revised draft.
Comment 3:
Line 123 states that Figure 3 contains data on CYP450 isoforms inhibition but no such data shown.
Response 3:
Thanks for your useful advice that made this paper much more readable and understandable. I have also included the inhibition parameter in all plots within Figure 3. The Figure 3 was shown as below:
|
|
Figure 3. Inhibition curves of lekethromycin for canine CYP450 activities using probe substrates in dog liver microsomes, phenacetin-O-deethylation for CYP1A2, 5-hydroxylation coumarin for CYP2A6, 1-hydroxylation bupropion for CYP2B6, 4-hydroxylation tol-butamide for CYP2C9, dextromethorphan O-demethylation for CYP2D6, 1-hydroxylation chlorzoxazone for CYP2E1, and 6β-hydroxylation testosterone for CYP3A4. The data was plotted as the mean ± standard deviation (SD) (n=3).
Comment 4:
Figure 4 requires improvement – is almost impossible to discern which plot corresponds to which legend line.
Response 4:
Thanks for your useful advice that made this paper much more readable and understandable. Figure 4 has been enhanced and changed, allowing for a clearer presentation of the results. The Figure 4 was showed as bellows:
Figure 4. Primary Lineweaver-Burk plot, the secondary plot for Ki and αKi in dog liver microsomes. Each point is represented as mean ± standard deviation (SD) (n=2).
Comment 5:
In the beginning of Abstract – “Lekethromycin (LKMS) is a derivative synthetic macrolide compound with an excellent 13 pharmacological property of antibacterial.” – no reference is provided.
Response 5:
Thanks for your useful advice. References have been included to introduce LKMS in the introduction. Additionally, the structural formula of LKMS will be provided in the Figure S3 as support data. Furthermore, some raw data will be made if necessary. The references were shown as below:
- P. Sun et al., “Plasma Protein Binding Rate and Pharmacokinetics of Lekethromycin in Rats,” Antibiotics, vol. 11, no. 9, p. 1241, Sep. 2022, doi: 10.3390/antibiotics11091241.
- Y. Cao et al., “Determination of lekethromycin in plasma and tissues of pneumonia-infected rats by ultra-high performance liquid chromatography-tandem mass spectrometry,” Journal of Chromatography B, vol. 1227, p. 123811, Jul. 2023, doi: 10.1016/j.jchromb.2023.123811.
Comment 6:
Introduction, line 38 – “And the result was observed in dogs in the further study in our lab.” – no reference is provided.
Response 6:
Thanks for your useful advice. The results of the relevant research on LKMS in dogs have not been published, but the in vivo results align with the inhibition observed in dog liver microsomes. To enhance clarity, the sentence has been removed.

Round 2
Reviewer 3 Report
Comments and Suggestions for Authors
Generally, authors have answered my previous comments.
Additional points - authors should improve Figure 5 in same way as Figure 4 for better presentation. Also, while readability of manuscript was enchanced, there is still room for improvement so authors should recheck manuscript for language mistakes.
Some examples: "Due to the complexity of diseases and drug resistance, combining chemical drugs was widely used to treat infectious diseases." should be "Due to the complexity of diseases and drug resistance, combining chemical drugs is widely used to treat infectious diseases."
"The results indicated that LKMS was a potent inhibitor of CYP2A6, CYP2B6, and CYP2D6 with IC50 values of 62.53 μΜ, 83.29 μΜ, and 39.66 μΜ, respectively..." should be "The results indicated that LKMS is a potent inhibitor of CYP2A6, CYP2B6, and CYP2D6 with IC50 values of 62.53 μΜ, 83.29 μΜ, and 39.66 μΜ, respectively..."
Comments on the Quality of English Language
While readability of manuscript was enchanced, there is still room for improvement so authors should recheck manuscript for language mistakes.
Some examples: "Due to the complexity of diseases and drug resistance, combining chemical drugs was widely used to treat infectious diseases." should be "Due to the complexity of diseases and drug resistance, combining chemical drugs is widely used to treat infectious diseases."
"The results indicated that LKMS was a potent inhibitor of CYP2A6, CYP2B6, and CYP2D6 with IC50 values of 62.53 μΜ, 83.29 μΜ, and 39.66 μΜ, respectively..." should be "The results indicated that LKMS is a potent inhibitor of CYP2A6, CYP2B6, and CYP2D6 with IC50 values of 62.53 μΜ, 83.29 μΜ, and 39.66 μΜ, respectively..."
Author Response
Comment 1:
Additional points - authors should improve Figure 5 in same way as Figure 4 for better presentation.
Response 1:
Thanks for your useful advice that make this paper much more readable and understandable. Figure 5 has been revised and improved to provide a clearer presentation of the result. The Figure 5 is showed as below:
Figure 5. The IC50 shift curve of lekethromycin in dog liver microsomes, 5-hydroxylation coumarin for CYP2A6, 1-hydroxylation bupropion for CYP2B6, dextromethorpham O-demethylation for CYP2D6. Each point represents the mean ± standard deviation (SD) (n=3).
Comment 2:
Also, while readability of manuscript was enchanced, there is still room for improvement so authors should recheck manuscript for language mistakes.
Some examples: "Due to the complexity of diseases and drug resistance, combining chemical drugs was widely used to treat infectious diseases." should be "Due to the complexity of diseases and drug resistance, combining chemical drugs is widely used to treat infectious diseases."
"The results indicated that LKMS was a potent inhibitor of CYP2A6, CYP2B6, and CYP2D6 with IC50 values of 62.53 μΜ, 83.29 μΜ, and 39.66 μΜ, respectively..." should be "The results indicated that LKMS is a potent inhibitor of CYP2A6, CYP2B6, and CYP2D6 with IC50 values of 62.53 μΜ, 83.29 μΜ, and 39.66 μΜ, respectively..."
Response 2:
Thanks for your useful advice. The grammar of this article has been thoroughly rectified and refined throughout of the text. The paper has undergone meticulous scrutiny, including proofreading and further enhancement by a native English speaker.
